# Phenotypical Flexibility of the EGFRvIII-Positive Glioblastoma Cell Line and the Multidirectional Influence of TGFβ and EGF on These Cells—EGFRvIII Appears as a Weak Oncogene

**DOI:** 10.3390/ijms232012129

**Published:** 2022-10-12

**Authors:** Aneta Włodarczyk, Cezary Tręda, Adrianna Rutkowska, Dagmara Grot, Weronika Dobrewa, Amelia Kierasińska, Marta Węgierska, Tomasz Wasiak, Tadeusz Strózik, Piotr Rieske, Ewelina Stoczyńska-Fidelus

**Affiliations:** 1Department of Tumor Biology, Medical University of Lodz, Zeligowskiego 7/9 St., 90-752 Lodz, Poland; 2Department of Research and Development, Celther Polska LTD, Inwestycyjna 7 St., 95-050 Konstantynow Lodzki, Poland; 3Department of Research and Development, Personather LTD, Inwestycyjna 7 St., 95-050 Konstantynow Lodzki, Poland; 4Department of Molecular Biology, Medical University of Lodz, Zeligowskiego 7/9 St., 90-752 Lodz, Poland

**Keywords:** apoptosis, EGFRvIII, EMT-like phenomenon, GB, TGFβ, senescence

## Abstract

Background: The biological role of EGFRvIII (epidermal growth factor receptor variant three) remains unclear. Methods: Three glioblastoma DK-MG sublines were tested with EGF (epidermal growth factor) and TGFβ (transforming growth factor β). Sublines were characterized by an increased percentage of EGFRvIII-positive cells and doubling time (DK-MG^low^ to DK-MG^extra-high^), number of amplicons, and EGFRvIII mRNA expression. The influence of the growth factors on primary EGFRvIII positive glioblastomas was assessed. Results: The overexpression of exoEGFRvIII in DK-MG^high^ did not convert them into DK-MG^extra-high^, and this overexpression did not change DK-MG^low^ to DK-MG^high^; however, the overexpression of RAS^G12V^ increased the proliferation of DK-MG^low^. Moreover, the highest EGFRvIII phosphorylation in DK-MG^extra-high^ did not cause relevant AKT (known as protein kinase B) and ERK (extracellular signal-regulated kinase) activation. Further analyses indicate that TGFβ is able to induce apoptosis of DK-MG^high^ cells. This subline was able to convert to DK-MG^extra-high^, which appeared resistant to this proapoptotic effect. EGF acted as a pro-survival factor and stimulated proliferation; however, simultaneous senescence induction in DK-MG^extra-high^ cells was ambiguous. Primary EGFRvIII positive (and SOX2 (SRY-Box Transcription Factor 2) positive or SOX2 negative) glioblastoma cells differentially responded to EGF and TGFβ. Conclusions: The roles of TGFβ and EGF in the EGFRvIII context remain unclear. EGFRvIII appears as a weak oncogene and not a marker of GSC (glioma stem cells). Hence, it may not be a proper target for CAR-T (chimeric antigen receptor T cells).

## 1. Introduction

Despite many years of analysis, the biological role and molecular mechanism of action of EGFRvIII remain unexplained. It is very important to recognize the complicated responses of EGFRvIII-positive cells to different environments, especially since EGFRvIII is regarded as an ideal (in terms of cancer uniqueness—a specific neo-antigen) target for immunotherapies, such as chimeric antigen receptor T (CAR-T) [1].

The EGFRvIII gene (epidermal growth factor receptor variant three) consists of exon 1 and exons 8–28 of EGFRwt (epidermal growth factor receptor wild type). It arises as a consequence of deletions within EGFR, leading to a lack of exons 2–7 of EGFR in the mRNA (deletion of aa 6–273) [2]. Moreover, combining exon 1 and exon 8 results in the presence of new glycine in the junction site. Hence, EGFRvIII is a very interesting target for immunotherapies such as CAR-T [3].

Importantly, EGFRvIII is usually present in amplicons. This type of mutation, with a possible new function and high number of gene copies, suggests that EGFRvIII is a strong oncogene. The lack of an EGF (epidermal growth factor)-binding domain suggests the constitutive activation of the receptor. However, the biological function of EGFRvIII remains enigmatic. Fan et al. suggest that EGFRvIII is activated only in the presence of EGF and that EGFRvIII heterodimerizes with EGF-activated EGFRwt [4]. Our previous data do not support this hypothesis: EGFRvIII-EGFRvIII homodimerization was observed, including the covalent type, with EGF having no influence on EGFRvIII phosphorylation. Nevertheless, the kinase activity of EGFRvIII is rather weak [5]. The lack of obvious membrane functions led some authors to analyze the nuclear role of EGFRvIII, with unclear findings [6]. It is more difficult to define the role of *EGFRvIII* since it is expressed in only one commercial cell line (DK-MG). Efforts to stabilize more cell lines are usually unsuccessful.

EGFRvIII has been proposed to take part in strong signal transduction, reprogramming ability, and interaction with transcription factors [7,8,9]. However, the influence of EGFRvIII on glioblastoma cells remains unclear [10,11,12]. This question is especially important if CAR-T therapy is considered. It would be more difficult for glioblastoma (GB) cells to downregulate the expression of a protein that plays a key role in gliomagenesis. Nevertheless, the percentage of EGFRvIII-positive cells in glioblastoma is usually lower than 30% [13,14]. Although this undermines the importance of this protein, some authors claim that the entire tumor depends on this cell subpopulation [15,16,17] or even cancer stem cells [8,18,19,20]. As such, the consequences of eliminating minority cell populations are difficult to predict. A suitable study model is needed. Hence, there is an pressing need to understand the influence of cytokines and growth factors (important in immunological networks) on glioblastoma EGFRvIII-positive cells.

Transforming growth factor β (TGFβ) has long been known to have a diverse impact on neoplastic cells [21]. However, in the glioblastoma environment, it has been mostly demonstrated as a stimulator of proliferation, a tumor expansion factor, and a trigger of epithelial to mesenchymal transition (EMT) [22]. Only single reports suggested that it may have anticancer effects, especially in regard to GB stem cells [23]. Although EMT and apoptosis seem to be mutually exclusive, both may occur as interrelated TGFβ-mediated events [21]. Importantly, EMT can be associated with tumor cell resistance, e.g., to tyrosine kinase inhibitors (TKIs), or with the inhibition of tumor growth [24]. Therefore, TGFβ is an interesting cytokine due to its impact on various cellular processes as well as the characteristic mechanism of its activation.

Similarly to TGFβ, EGF also plays an important role in glioblastoma biology. Some authors suggested that *EGFRvIII* is not a fully independent oncogene but a protein heterodimerizing with EGFR activated by EGF [4,25]. Such models do not offer an easy explanation for observations suggesting that EGF has an antiproliferative role during primary EGFR/EGFRvIII-amplified glioblastoma cell culture [26].

In vitro senescence in glioblastoma cells is another important phenomenon. EGF can be crucial, considering that oncogene-induced senescence is one of several types of senescence. Our results suggest that primary GB cells became senescent in vitro more easily than pericytes. During these experiments, all media contained EGF [27,28].

The findings regarding the biological role of EGFRvIII appear incoherent. This may be associated with the lack of an appropriate in vitro model, as cell lines endogenously expressing EGFRvIII are extremely rare, probably due to their in vitro prosenescence [27,29]. However, the DK-MG cell line seems to offer unique features and extreme flexibility during in vitro culture.

The present study attempts to determine the role of TGFβ and EGF in glioblastoma EGFRvIII-positive cells based on three DK-MG sublines with different EGFRvIII expression status: DK-MG^low^, DK-MG^high^, and DK-MG^extra-high^. Such models perfectly mimic the natural heterogeneity of glioblastoma cells and reflect the different tumor development stages and malignancy. They also analyze the impact of examined compounds on the primary GB cell cultures as the most relevant model of natural conditions.

Our data indicate that the effects of TGFβ and EGF on EGFRvIII-positive neoplastic cells are more complicated than previously assumed. This can be especially important due to the constant lack of effective therapy against GB and the fact that patient survival is typically less than 11–13 months from diagnosis, even with the currently most effective therapeutic regimen [30].

## 2. Results

### 2.1. DK-MG Subline Characterization

Three DK-MG sublines were characterized: DK-MG^low^, DK-MG^high^, and DK-MG^extra-high^. Immunocytochemical analysis showed that the DK-MG^high^ cells demonstrated both EGFR^vIII^-positive and EGFR^vIII^-negative cells (Figure 1A). The percentage of EGFR^vIII^-positive cells in DK-MG^high^ varied from 40 to 60%, calculated based on immunocytochemical analyses with anti-EGFR^vIII^-specific antibodies. In the DK-MG^low^ subline, only single cells with EGFR^vIII^ expression were observed, with an estimated percentage of less than 5% of the whole population, whereas 100% of DK-MG^extra-high^ cells were EGFR^vIII^ positive (Figure 1A). DK-MG cells with higher EGFR^vIII^ expression were characterized by a neuroepithelial phenotype and enhanced proliferation, while cells with low EGFR^vIII^ expression by the mesenchymal phenotype and slow proliferation and/or spontaneous apoptosis in both serum and serum-free media (Figure 1A and Appendix A) [5]. Despite using next-generation sequencing, no important genomic differences were observed between DK-MG sublines (Table 1) other than numbers of EGFR^vIII^ amplicons [5].

The DK-MG^extra-high^ subline showed very high EGFR^vIII^ protein expression and this level was considerably higher than that in DK-MG^high^ (Figure 1B). In the case of DK-MG^low^, EGFR^vIII^ was practically undetectable in Western blot analysis. Compared to the protein level, the mRNA level was 1.2 times higher in DK-MG^extra-high^ than in DK-MG^high^ but was only present in small amounts in DK-MG^low^ cells (Figure 1B,C). A decrease in EGFRwt upon EGF stimulation was observed in each subline as an effect of the ligand inducing EGFR internalization and subsequent lysosomal degradation of activated EGFR: a typical mechanism of non-mutated EGFR signaling termination (Figure 1B). The MLPA (Multiplex-ligation dependent probe amplification) analysis of DK-MG^extra-high^, DK-MG^high^, and DK-MG^low^ did not detect any significant alterations in the gene profile beyond the differences in EGFR gain [D]. The MLPA analysis, based on calculating the ratio between the normalized peak heights of samples and controls for each EGFR probe, of DK-MG^extra-high^ ranged between 16 and 20 (p105 set) for EGFR^vIII^ amplicons and between 12 and 14 for DK-MG^high^ and 1 and 2 for DK-MG^low^ (p105 set) (Appendix A).

The overexpression of exogenous (genetically engineered) EGFR^vIII^ did not change the phenotype of DK-MG^high^ into DK-MG^extra-high^, and the overexpression in DK-MG^low^ did not result in a change into the DK-MG^high^ or DK-MG^extra-high^ phenotype regardless of the cell culture conditions (Figure 2A–D). In general, the doubling time did not change in these cells following exogenous EGFR^vIII^ overexpression (Table 2). A control rescue experiment in DK-MG^low^ was performed by the overexpression of K-RAS^G12V^ (Kirsten rat sarcoma viral oncogene homolog) from an exogenously-introduced transgene. In contrast to EGFR^vIII^, the DK-MG^low^ cells returned to DK-MG^high^ cells following treatment with mutated K-RAS (glycine to valine in codon 12 of K-RAS) in a dose-dependent manner (Figure 2E). Western blot analyses confirmed the activation of ERK; and AKT in these conditions.

DK-MG^low^ showed lower levels of phospho EGFR^vIII^ than DK-MG^high^; however, DK-MG^high^ demonstrated similar exogenous overexpression of (DK-MG^low exovIII^) EGFR^vIII^ and phospho EGFR^vIII^. It should be noted that while the EGFR^vIII^ protein was detected in about 40–60% of DK-MG^high^ cells, exogenous expression generated 100% of EGFR^vIII^-positive cells, which can significantly affect the observed results. Although the DK-MG^extra-high^ subline demonstrated very high levels of EGFR^vIII^ phosphorylation, ERK and AKT phosphorylation was lower than expected (Figure 2F). Finally, the exogenous expression of EGFR^vIII^ in DK-MG^low^ was insufficient for avoiding spontaneous cell death (Figure 2G).

### 2.2. EGF Influence on DK-MG Sublines

Our findings highlight the two-directional role of EGF in the DK-MG^high^ subline. Firstly, EGF increased the number of proliferating cells: 96-h incubation of the DK-MG^high^ cell line with the epidermal growth factor induced a significant increase in the percentage of BrdU (Bromodeoxyuridine)-positive EGFR^vIII^-negative cells and only a slight increase in the number of BrdU-positive EGFR^vIII^-positive cells. Moreover, the percentage of proliferating EGFR^vIII^-negative cells significantly exceeded the percentage of BrdU-positive EGFR^vIII^-positive cells, which was confirmed by statistical analysis (Figure 3A,B). The addition of EGF to the DK-MG^high^ cell culture also significantly promoted a decrease in the percentage of apoptotic cells after both 72 and 96 h of incubation (Figure 3C–E). Although EGF may be a prosurvival factor, the results of our study indicate that this factor induced an elevation in SA-β-Gal (Senescence-associated beta-galactosidase)-positive DK-MG^high^ cell numbers. EGF contributed to a slight increase in the percentage of senescent EGFR^vIII^-negative cells after 96-h incubation, while the opposite tendency was noted for SA-β-Gal-positive EGFR^vIII^-positive cells (Figure 3F). Both observed dependencies were not statistically significant. Very few BrdU-positive cells were also SA-β-Gal positive, and this number did not change in response to EGF treatment. Moreover, the addition of EGF to the cell culture induced a significant increase in the percentage of EGFR^vIII^-negative DK-MG^high^ cells (Figure 3G). The DK-MG^extra-high^ subline did not contain a senescent subpopulation: only single cells were senescent, both in the presence or absence of EGF (Appendix A). DK-MG^extra-high^ cells also demonstrated only single apoptotic cells in several tested in vitro conditions (Appendix A). The DK-MG^low^ subline was in general very prone to senescence and showed a high percentage of apoptotic cells, especially in serum-free conditions (Appendix A).

### 2.3. TGFβ Influence on DK-MG Sublines

Long-term cell microscopic observation (60 h of incubation with cytokine) of DK-MG sublines found TGFβ to have different effects depending on cell type, ranging from proliferation and apoptosis to epithelial-to-mesenchymal transition (EMT). A cytostatic effect was observed in DK-MG^high^ cells (Figure 4A) whereas no difference was observed in DK-MG^low^ (Appendix A). Moreover, this effect was also not observed in other cells treated with TGFβ, such as NIH/3T3 fibroblasts (Figure 4B). In the case of iNS (induced neural stem cells), statistically insignificant differences were observed (*p* = 0.085) (Appendix A). Caspase assay indicated that the number of apoptotic cells increased in such conditions between the control (left photo) and tested group (right photo) (Figure 4C). Parallel performed real-time microscopic observations indicated that implemented culture conditions (serum starvation) were not harmful for cells (left photo); however, in the case of TGFβ treatment (right photo), only single, viable cells remained in culture, with characteristic mesenchymal-like morphology specific to the EGFR^vIII^-negative or oncogene low-expressed DK-MG subpopulation (Figure 4D). The results calculated from five independent experimental panels showed the caspase assay results to be statistically significant (Figure 4E); in addition, the percentage of viable cells decreased after TGFβ treatment (Figure 4F).

These relationships prompted a more precise analysis of the impact of TGFβ on different cell types. The following IC_50_ (half-maximal inhibitory concentration) values were determined: DK-MG^high^ was 4 ng/mL, NIH/3T3 was 1000 ng/mL, and all other cell types were >1000 ng/mL (Table 3). The results presented for TGFβ1 are representative for all three isoforms.

### 2.4. Characteristics and Influence of Analyzed Growth Factors on Primary EGFRvIII-Positive Glioblastomas

Analyses of EGF and TGFβ were performed on different cellular models that better reflect in vivo conditions. An EGFRvIII positive primary GB cell line was obtained and stabilized; EGFRvIII status was confirmed at the protein level by immunofluorescence and Western blotting (Figure 5A,B) and at the mRNA level by qRT-PCR (Real-Time Quantitative Reverse Transcription PCR) (Figure 5C). To confirm whether EGFRvIII is a marker of GB stem cells, EGFRvIII and SOX-2 protein co-staining was performed in three cases of EGFRvIII-positive GB. Both EGFRvIII-positive SOX-2-positive and EGFRvIII-positive SOX-2-negative cells were observed (Figure 5D), which seems to undermine the role of SOX-2 as a marker of these cells. MLPA analysis of EGFRvIII gain was also performed in primary GB cultures, similar to the stable DK-MG cells, but as EGFRwt amplification was observed in most EGFRvIII-positive cases, the MLPA results were not interpretable in primary cultures (Figure 5E).

A differential effect of EGF on EGFRvIII-positive glioblastoma cells was observed. Even spheres isolated from the same EGFRvIII-positive primary GB can respond to EGF in opposite ways. Just after the 3D glioblastoma structures were transferred into adherent conditions (Figure 5F), some proliferated and others deteriorated. In our previous studies, the longest cultivation times of GB cultures were achieved by culturing GB cells under 3D conditions in EGF-free medium. However, even such culture conditions did not prevent senescence among GB cells [28].

In the case of TGFβ, we also observed a variable effect on cells isolated from GB EGFRvIII. In one case, the IC50 was measurable at 14 ng/mL (Table 3). This value was several times higher than in the case of DK-MG^high^. In the other two cases, the IC50 was undetectable (no cytotoxic effect was observed, Table 3). Importantly, TGFβ was found to have an effect in cultures with an exceptionally high percentage of EGFRvIII cells (75%); however, DK-MG^extra high^ demonstrating 100% EGFRvIII-positive cells did not respond to TGFβ. In all analyzed cases, TGFβ1-3 triggered the selection of mesenchymal-like cells or transition to mesenchymal cells (Figure 5G). The cell number continued to increase during 60-h incubation with cytokines (Figure 5H).

## 3. Discussion

Our work examines the influence of EGF and TGFβ on DK-MG sublines. DK-MG appeared to be very flexible cell line with three sublines. One of them showed almost undetectable EGFRvIII mRNA expression and a very slow proliferation rate (DK-MG^low^). The second (DK-MG^high^) presented two populations of cells, i.e., with and without EGFRvIII protein, and a moderate proliferation rate. A third cell subline expressed high EGFRvIII expression for all cells (DK-MG^extra-high^) and the shortest doubling time. The number of amplicons and mRNA expression were similar in the DK-MG^extra-high^ and DK-MG^high^ sublines. DK-MG^extra-high^ cells demonstrated the highest level of phospho EGFRvIII. Importantly, the observed DK-MG sublines did not differ in terms of essential DNA sequence changes in NGS analyses.

Our findings indicate that TGFβ is able to trigger the apoptosis of DK-MG^high^ cells. It must be admitted that the DK-MG^low^ and DK-MG^high^ cell sublines showed apoptotic cells in many tested conditions, but the cell culture remained stable (number of proliferating cells prevailed). TGFβ increased the number of apoptotic cells in the DK-MG^high^ subline [31]. The results of analyses on the DK-MG^high^ subline were to some extent confirmed during analyses of EGFRvIII-positive primary GB cells. Following exposure to TGFβ, primary GB cells without mesenchymal morphology were gradually eliminated. In the presence of TGFβ, primary GB cells were overgrown by normal cells that mostly consisted of pericytes and other glioblastoma-associated stromal cells. In general, any analysis of primary glioblastoma cells is very difficult, and these cells showed a tendency to become senescent in various cell culture conditions, as discussed below. As demonstrated by IC50 values, pericytes are insensitive to TGFβ, which can explain the slow and gradual selection of normal versus primary GB cells in these conditions. The TGFβ-induced EMT-like phenomenon observed in the case of DK-MG^high^ cells cannot be considered a culprit of classical therapeutic resistance of GB cells to inter alia TKIs (tyrosine kinase inhibitors). Following TGFβ-induced EMT-like processes, the DK-MG cells proliferated very slowly or even seemed to undergo senescence.

In contrast, the DK-MG^extra-high^ cells were resistant to TGFβ. This growth factor was unable to induce apoptosis in these cells; alternatively, the rapid proliferation of these cells makes this process negligible.

The second analyzed growth factor, EGF, also demonstrated a complex influence on DK-MG cells. DK-MG^high^ consisted of two cell fractions (EGFRvIII positive and negative) and presented a stable percentage of EGFRvIII-negative and -positive senescent cells after exposure to EGF. Both subpopulations (EGFRvIII positive and negative) exposed to EGF demonstrated an increased percentage of BrdU positive cells. Surprisingly, in spite of a stable percentage of SA-β-Gal-positive cells and the proliferation of both subpopulations (EGFRvIII positive/negative), a decrease in the percentage of EGFRvIII-positive cells was observed. This suggests that either some EGFRvIII-positive cells lost EGFRvIII expression during exposure to EGF or became apoptotic. The percentage of apoptotic cells decreased in the DK-MG^extra-high^ subline after exposure to EGF. This favors a switch from EGFRvIII-positive to EGFRvIII-negative cells instead of apoptosis of EGFRvIII-positive cells; hence, the percentage of EGFRvIII-positive senescent cells was slightly decreased and that of EGFRvIII-negative cells slightly increased after DK-MG^high^ exposure to EGF. Although the changes were statistically insignificant, this supports the hypothesis of a gradual decrease in EGFRvIII expression in senescent cells. Combining these data may suggest that EGF has a very complicated and divergent influence on this subline. Similar data were obtained for the primary glioblastoma cell cultures. EGF was already suggested as a factor negatively selecting primary EGFRvIII-positive GB cells in vitro in long-term analyses.

The senescence of DK-MG cells is intriguing since these cells show mutation of TP53 and homozygous deletion of CKDN2a (cyclin-dependent kinase inhibitor 2A). Both proteins have often been described as playing a key role in senescence. One could predict oncogene-induced senescence in an EGFRvIII-positive population. However, in the present study, although the DK-MG^high^ subline showed senescent cells both before and after exposure to EGF (serum-free conditions), many cells exposed to EGF continued to proliferate (confirmed by BrdU analysis). BrdU incorporation and SA-β-Gal activity were almost mutually exclusive (96 h of BrdU incorporation), excluding a quick shift from proliferation to senescence.

The analysis of transduction pathways did not clarify the mechanisms behind these discrepancies. EGFRvIII was highly phosphorylated in DK-MG^extra-high^ cells, but interestingly, EGFRvIII did not have a strong influence on the AKT and ERK pathways. Moreover, EGF did not appear to influence EGFRvIII activation.

The DK-MG^extra-high^ subline did not contain large senescent or apoptotic subpopulations: only single senescent cells were observed in the presence or absence of EGF and TGFβ. The DK-MG^low^ subline showed the highest percentage of apoptotic cells, especially in serum-free conditions.

Generally, the literature defines EGF and TGFβ as proneoplastic at advanced stages of glioblastoma tumorigenesis [22,32]. Our data suggest that this model is oversimplified. In fact, these factors cause different responses among glioblastoma cells. It is possible that GB cells are very flexible in constantly changing environments. Recent studies suggest that changes in amplicon numbers and sets may be responsible for these different responses and the environmental flexibility and resistance of GB cells [5]. However, our data indicate no significant differences between DK-MG^high^ and DK-MG^extra-high^ sublines with regard to gene copy numbers, including EGFRvIII. Similarly, NGS analysis did not highlight any important genetic differences between sublines. EGFRvIII protein expression seems to influence the final response of DK-MG cells to various factors such as EGF or TGFβ, but it is also possible that another factor plays an important and independent role in regulating the final outcome.

DK-MG^low^ cells and the DK-MG^extra-high^ subline were generated several times. No reversion was observed from the DK-MG^low^ to DK-MG^high^ phenotype; however, reversion was found from DK-MG^extra-high^ to DK-MG^high^. Unfortunately, since EGFRvIII is amplified, it cannot be knocked-out using CRISPR (Clustered Regularly Interspaced Short Palindromic Repeats) technology. In DK-MG cell lines, the number of EGFRvIII copies per cell tends to be in the thousands [33]. Our previous efforts to use siRNA were limited since shRNA dedicated specifically to EGFRvIII could not be designed, and shRNA downregulated both EGFRvIII and EGFRwt mRNA [34].

The present study examined whether EGFRvIII transgene overexpression can change the DK-MG cell phenotype from low to high/extra-high or from high to extra-high. Surprisingly, the DK-MG^low^ subline with overexpression of EGFRvIII did not become similar to DK-MG^extra-high^, or even the DK-MG^high^ subline, whereas the DK-MG^high^ cell line, showing the expression of EGFRvIII in all cells, did not become similar to the DK-MG^extra-high^. No changes in the aggressiveness of DK-MG^low^ cells were observed, even though the level of EGFRvIII and phospho EGFRvIII observed after EGFRvIII transgene overexpression was comparable to the level observed in DK-MG^high^. EGFRvIII phosphorylation was lower in DK-MG^high^ cells showing overexpression of EGFRvIII than in the DK-MG^extra-high^ subline cells. Intriguingly, in that context we did not observe discrepancies between different sublines in terms of AKT and ERK phosphorylation corresponding to the very high phosphorylation level of EGFRvIII observed in the DK-MG^extra-high^ subline. The DK-MG^extra-high^ subline showed higher AKT and ERK phosphorylation than other sublines, but was significantly different to the EGFRvIII phosphorylation level. The EGFRvIII influence on the STAT3 (signal transducer and activator of transcription 3) transcription factor was proposed as an alternative for AKT/ERK-dependent transduction regulation [35]. All these findings, and previous analyses [34], suggest that the influence of EGFRvIII on cell activity depends on the complicated molecular context and actions of other proteins. It is possible that EGFRvIII is an epiphenomenon and other factors are responsible for changes in DK-MG sublines phenotypes. The analysis of *RASG12V* overexpression in DK-MG supports this suggestion since this oncogene caused DK-MG^low^ rescue (phenotype change from DK-MG^low^ to a phenotype similar to DK-MG^high^). We have already described a lack of phenotypical changes in many cell lines overexpressing exogenous EGFRvIII [5].

Obviously, the DK-MG cell line is far from an optimal representation of EGFRvIII positive glioblastoma tumors. However, in spite of their unusual genotype (coexistence of EGFRvIII and TP53 mutations), these cells seem to be the best available representation of EGFRvIII-positive primary GB cell cultures. Firstly, EGFRvIII-positive neoplastic cell lines are unique, but DK-MG cell sublines also show many pitfalls typical for primary cell cultures including in vitro growth instability and phenotypical changes coherent with primary GB cell flexibility. DK-MG^high^ cells resemble primary glioblastoma cells, consisting of EGFRvIII positive and negative cells. Besides oncological issues, the DK-MG cell line seems to represent a borderline example of cell line flexibility, requiring unusual attention for recognizing this cell line status (subtype). DK-MG cells demonstrate a unique ability to evolve in vitro; this is an interesting model, but offers additional challenges during experiments.

It should be considered that many differences exist between in vitro and in vivo conditions, and even DK-MG cells are far from primary GB cells. Moreover, some environmental changes are difficult to accomplish in vivo. For example, in physiological conditions, TGFβ is present in the form of an inactive precursor further activated by various mechanisms [36]. In vitro conditions allow the concentrations of this growth factor to become very high.

We do not suggest that TGFβ or EGF should be hastily considered therapies for any form of GB. Many articles have shown these factors to have a positive influence on GB growth and survival. Moreover, as we noticed, the final outcome could be far from therapeutic in the tumor environment, as the DK-MG^high^ cell line evolved to DK-MG^extra high^, showing a high expression of EGFRvIII in all cells and a lack of response to TGFβ and EGF. EGFRvIII-positive primary GB cells demonstrated a differential response to EGF and TGFβ. Moreover, this effect could differ between cells isolated from a single tumor. Undoubtedly, further studies are needed to explain this different response.

Many years of our research on glioblastoma biology indicate that EGF is not able to prevent GB cell senescence in vitro [28]. It seems that the longest proliferation time and original genotype maintenance can be observed when the cells are grown under 3D conditions without EGF [28]. Under these conditions, however, no robust increase in the cell number occurs. Under adherent conditions, cell number increases faster, but so does the loss of the original genotype. It cannot be excluded that fast proliferation contributes to it, via the DNA damage response phenomenon characterized by a temporary, rapid increase in proliferation with subsequent senescence.

EGFRvIII-positive cells seems to be an ideal target for CAR-T [1]. However, the presence of EGFRvIII-negative and -positive cells in a single tumor is a very intriguing phenomenon typical of glioblastoma and is represented by the DK-MG^high^ subline. Importantly, the elimination of one population of cells by CAR-T therapy may not lead to a satisfying outcome, but EGFRvIII can still be defended as a target for CAR, assuming that it is marker of GB stem cells. In this scenario, the elimination of even a small fraction of cells would lead to whole tumor eradication.

The concept of GB stem cells is controversial [37]. Our research indicates that SOX-2 rather than CD-133 [37] is a marker of cells derived from phenotypically-different GB cells. Due to the hypothesis that EGFRvIII is a marker of stem cells in glioblastomas or even in other cancers [8], EGFRvIII SOX-2 co-staining was conducted to address this question. This is especially important since even a small percentage of EGFRvIII-positive cells, if EGFRvIII is expressed on the surface of GB stem cells, may be a good target for CAR-T. If a small portion of the EGFRvIII-positive GB cells are not GB stem cells then this target is meaningless. We observed both EGFRvIII-positive SOX-2-positive and EGFRvIII-positive SOX-2-negative cells. This undermines the role of EGFRvIII as a protein characteristic of GB stem cells, but EGFRvIII can still be recognized by a synthetic receptor such as synNOTCH, triggering the local expression of CARs targeting unspecific membrane molecules such as EphA2 (ephrin type-A receptor 2) or IL13Rα2 (interleukin 13 receptor alpha 2) [38]. Nevertheless, several articles showed that primary EGFRvIII-positive glioblastoma recurrences can be EGFRvIII negative [39,40]. To this end, we suggest that EGFRvIII should be considered important in the early stages of glioblastoma genesis.

In conclusion, the effects of TGF and EGF on EGFRvIII-positive GB cells are diverse. EGF can cause both proliferation and senescence. TGFβ may not have a pronounced effect through epithelial to mesenchymal transition up to induction of apoptosis. The role of EGFRvIII seems to be overestimated so far and without any obvious effect on the signal transduction pathway. Unlike RAS^G12V^, it does not allow a rescue effect. It also does not seem to be an important protein for glioblastoma stem cells. Further studies are therefore needed before it can be confirmed as a suitable target for CAR-T.

## 4. Materials and Methods

### 4.1. Cell Cultures

The following cells were used: DK-MG, U87-MG, T98G, NIH/3T3 (all ATCC), primary GB cultures (for which tissue samples were obtained from patients diagnosed with glioblastoma and treated at the Clinical Department of Neurosurgery, The Voivodal Specialistic Hospital in Olsztyn), neural stem cells (NSC; Gibco, Thermo Fisher Scientific, Waltham, MA, USA), and pericytes (Celther Polska, Poland). All GB samples were collected using the protocol approved by the Bioethical Committee of the Medical University of Lodz (Approval No RNN/156/20/KE). Written informed consent was obtained from all patients and their data were processed and stored according to the principles expressed in the Declaration of Helsinki. The patients were diagnosed according to the World Health Organization Criteria for Brain Tumor Classification [41]. Stable lines were cultured according to the manufacturer’s recommendations, and primary GB cells according to the protocol by Xie et al. [42]. DK-MG was characterized as a heterogeneous cell line with populations positive and negative for EGFR^vIII^ expression [43]. Numerous clones with varying levels of EGFR^vIII^ expression were obtained by serial dilution, including clones with lower and higher levels of EGFR^vIII^, referred to as DK-MG^low^ and DK-MG^high^, respectively [5]. Subsequent serial dilution of DK-MG^high^ cells obtained DK-MG^extra-high^ cells used in the study. Moreover, cell lines with exogenous EGFR^vIII^ expression were obtained (DK-MG^low exovIII^ and DK-MG^high exovIII^).

### 4.2. Development of Cell Lines with Exogenous EGFR^vIII^ and K-RAS^G12V^ Expression

Preparation of EGFR^vIII^ inserts, lentivirus production, and the transduction procedure were performed as described previously [5,44]. Briefly, obtained lentivirus carrying EGFR^vIII^ was used in DK-MG^low^ and DK-MG^high^ transduction. Following clonal selection with puromycin, cells with exogenous EGFR^vIII^ expression were analyzed according to their viability, proliferation ratio, as well as morphology. In the same way, K-RAS^G12V^ lentiviruses were obtained, introduced into DK-MG cell sublines and analyzed.

### 4.3. Real-Time PCR

Real-time PCR analysis was performed as described previously [44]. Total RNA was isolated from both DK-MG cell sublines following endogenous and exogenous EGFR^vIII^ expression as well as from primary glioblastoma cell cultures at early passages. Efficiency was calculated using LinReg software and relative expression according to Pfaffl et al. [45].

### 4.4. EGFR Total DNA Copy Number Calculation

To determine the copy number of the EGFRvIII (calculated as the difference between EGFRwt and total EGFR) and EGFRwt genes, real-time PCR analyses at the DNA level were performed [27]. The RPP25 gene was used for normalization. Primer sequences used for the amplification of the EGFRwt gene at the DNA level were as follows: 5′-CTCACGCAGTTGGGCACTTT-3′, 5′-CCACCTCACAGTTATTGAACATCCT-3′ while 5′-CACACCCCTGACTCTCCACT-3′, 5′-GAGACAATCCTGTGAGCTTGG-3′ sequences were used to amplify the total EGFR gene. The following specific primers were used for the amplification of the RPP25 gene: 5′-GGGAGATGCGGAAGAATGT-3′, 5′-CCTCCAGTCAGCCACAGAA-3′. Results around 2 were interpreted as two copies of the gene 2, values below 2 were considered as deletion, and those above 2 as amplification.

### 4.5. Multiplex Ligation-Dependent Probe Amplification (MLPA)

MLPA was carried out using P105-D1 Glioblastoma-2 probemix (MRC-Holland, Cat. no. D1-0413) and SALSA MLPA EK1 kit-FAM (MRC-Holland, Cat. no. EK1-FAM) as described previously [28]. Thresholds used to define copy number variation were as recommended: deletion < 0.7, normal 1.3 > duplication.

### 4.6. Proliferation and Apoptosis Assessment

The proliferation ratio of DK-MG sublines (also with exogenous expression of EGFR^vIII^) was analyzed according to in vitro real-time observations in BioStation CT (Nikon, Tokyo, Japan) for 60 h in different cell culture conditions (serum free or 10% FBS media). Briefly, the cells were seeded onto 6-well plates in the amount of 40,000 cells per well and left to adhere for 24 h. After that, plates were placed in BioStation CT and photos were taken every 6 h. Simultaneously, an apoptosis analysis was conducted by adding CellEvent^TM^ Caspase-3/7 Green Ready Probes^TM^ Reagent (Thermo Fisher, Waltham, MA, USA) to the same wells and performing the observation under green fluorescent light for 96 h. The number of viable cells was calculated using CL Quant software (Nikon) from 5 visual fields. According to tools described above, the influences of 1–10 ng/mL TGFβ1 (Merck Millipore, Burlington, MA, USA), TGFβ2 (Sigma–Aldrich, St. Louis, MI, USA), TGFβ3 (Thermo Fisher Scientific, Waltham, MA, USA), and EGF (20 ng/mL, Peprotech, Waltham, MA, USA) on the proliferation ratio and apoptosis were analyzed on different cell lines: DK-MG sublines, NIH/3T3 and induced neural stem cells (iNSc). Erlotinib (10 µM, Molekula) and DMSO (vehicle, Serva) were used as controls.

### 4.7. Western Blotting

EGFRvIII and EGFRwt status in primary GB cell cultures, DK-MG cell sublines comparison as well as the analysis of TGFβ and EGF influence were performed by Western blotting and conducted as described previously [5,44]. The primary and secondary antibodies with dilutions are listed in Appendix A.

### 4.8. Immunocytochemical Staining

Immunocytochemical staining was performed as described previously [27]. Briefly, DK-MG^low^, DK-MGhigh, DK-MG^extra high^ as well as primary glioblastoma cells were seeded onto 4-well plates in the amount of 40,000 cells per well. After adhesion and 24-h starvation in medium without serum, EGF (20 ng/mL) was added to the half of the wells and incubated in serum-free conditions. After 96 h, the cells were washed three times with PBS, fixed (4% paraformaldehyde (PFA) in PBS, 14 min, RT), and permeabilized (0.1% Triton X-100, 10 min, RT). Nonspecific binding sites were blocked for 1 h with 2% donkey serum (Sigma Aldrich) in PBS. After that, fixed cells were incubated with the appropriate primary antibodies (1 h, RT, Appendix A) and further visualized by simultaneous incubation with a combination of species-specific fluorochrome-conjugated secondary antibodies (1 h, RT, Appendix A). Simultaneously, to eliminate an antibody fluorescent background effect, the control samples were incubated with the secondary antibodies alone and were subsequently processed as the analyzed cells. The slides were mounted with ProLong^®^ Gold Antifade Reagent with DAPI (Invitrogen), coverslipped, and examined using an OPTA-TECH MN 800 fluorescence microscope. The percentages of EGFR^vIII^-positive and negative cells were subsequently calculated from at least ten visual fields comprising at least 200 cells per case according to usage of ImageJ Software tools.

### 4.9. Senescence Associated (SA)-β-Galactosidase Assay

SA-β-Gal staining was performed using the Senescence Cells Histochemical Staining Kit (Sigma Aldrich) as described previously [27]. The cells were prepared for immunocytochemical staining. Briefly, after 96 h incubation with EGF, the cells were washed three times with PBS, fixed with cold 4% PFA for 10 min and then washed two times with PBS for 5 min. After that, the cells were incubated with freshly-prepared staining mixture (12 h, 37 °C, no CO_2_). Following incubation, cells were washed twice with PBS for 5 min and photographed using an Olympus CKX41 light microscope. The percentage of positively stained cells was subsequently calculated from at least ten visual fields comprising at least 200 cells per case.

### 4.10. Bromodeoxyuridine (BrdU) Incorporation Assay

The detection of proliferating cells was conducted as previously described [27]. Briefly, 10 μM BrdU and 20 ng/mL EGF were added to the cultures and after 96 h, immunocytochemical staining for other markers was performed as described above. Subsequently, cells were post-fixed in 4% PFA and permeabilized again with 0.1% Triton X-100 (10 min, RT). Non-specific binding sites were blocked by incubation with 2% donkey serum in PBS for 30 min. After blocking, the cells were treated with 2 N HCl (40 min, 37 °C) and then with 0.1 M borate buffer (pH  =  8.5, 12 min, RT). Then, the cells were incubated with anti-BrdU antibody (1:500, 1 h; Sigma–Aldrich), washed with PBS, and incubated with the appropriate secondary antibodies (1 h, RT in dark). Finally, the cells were mounted with ProLong^®^ Gold Antifade Reagent (Molecular Probes), coverslipped, and examined using an OPTA-TECH MN 800 fluorescence microscope. For each analysis, 200 nuclei were examined.

### 4.11. Next-Generation Sequencing (NGS)

Library preparation and sequencing were performed with Ion Torrent™ PGM (Thermo Fisher Scientific, Inc., Waltham, MA, USA) following the manufacturer’s protocol and as described previously [46], using whole genomic DNA isolated from DK-MG sublines.

### 4.12. IC_50_ Determination

For the purpose of IC_50_ determination, 3000 cells/well of 96-well plates were seeded, left overnight to adhere, and serum starved for 24 h. Various concentrations of TGFβ1 were added to the fresh serum-free medium (1–10 ng/mL range for DK-MG^high^ and primary GB cells and 100–1000 ng/mL for other cell types). After 72 h of incubation, the cells were washed twice with PBS (Gibco), incubated with 0.1% crystal violet solution (10 min; Merck Millipore), washed with PBS, and incubated with 10% glacial acetic acid (10 min; Sigma–Aldrich). Absorbance was measured at λ = 540 nm using a Synergy™ 2 Multi-Mode Microplate Reader (BioTek), enabling us to determine the IC_50_ value.

### 4.13. Statistical Analyses

Statistical significance for real-time PCR results was determined by two-way ANOVA with post-analysis Bonferroni’s multiple comparisons test. Statistical significance for Western blotting results was performed according to the densitometry results (measured in ImageJ software) obtained from three independent analysis and estimated by a paired Student’s t-test. The intensity of the fluorescence signal was measured from five independent experimental panels as a ratio of the integrated density to the area in ImageJ software and subsequently estimated by a paired Student’s t-test. The percentage of apoptotic, viable, senescent, and BrdU-positive cells was calculated from five independent experimental panels and estimated by a paired Student’s t-test. Proliferation curves were constructed according to the mean of five different points of view counted in CL-Quant software based on photos taken from 6-h intervals. All error bars on graphs and curves represented standard deviation values (SD). Statistical tests and graphical representative results were conducted and obtained with the use of GraphPad Prism software. In all analyses, the statistical significance of results was determined as follow: *p* ≤ 0.05 (*), *p* ≤ 0.01 (**) and *p* ≤ 0.001 (***).

## 5. Conclusions

EGFRvIII-positive cells demonstrated a multidirectional response to environmental changes. Firstly, treatment with TGFβ induced apoptosis. It also induced an EMT-like phenomenon in primary GB cells. On the other hand, incubation with EGF resulted in an increase in the proliferation ratio and cell viability, but it may also induce senescence. Both observations were surprising since both TGFβ and EGF mainly acted as pro-survival factors. It is also interesting that the exogeneous expression of the oncogene in particular DK-MG sublines seemed to be insufficient to create features corresponding to a subline with the appropriate expression of EGFRvIII. EGFRvIII primary GB cells responded differentially to TGFβ and EGF. EGFRvIII did not seem to be marker of glioblastoma stem cells. Targeting EGFRvIII-positive cells by CAR-T seems to make sense only in the context of synthetic receptors.

Hence, the role of EGFRvIII is more complex than previously assumed, and further research is needed. These studies should determine the exact role of this oncogene in biological processes to support the development of effective therapy against tumors with this type of mutation.

## Figures and Tables

**Figure 1 ijms-23-12129-f001:**
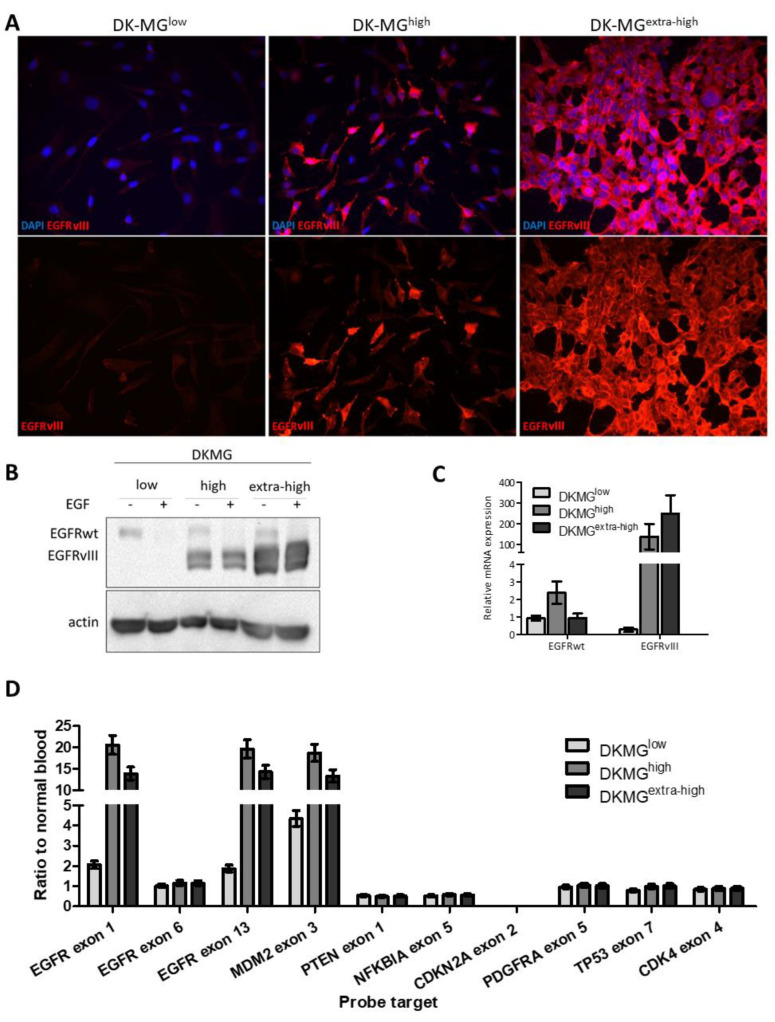
Comparison of DK-MG sublines. Different EGFR^vIII^ expressions were detected. In the DK-MG^low^ subline, this expression was negligible, but gradually increased accordingly for DK-MG^high^ (about 50% of EGFR^vIII^ –positive cells in the population) and DK-MG^extra-high^ (about 100% of EGFR^vIII^ –positive cells in the population) sublines. This relationship was observed on the protein level (immunocytochemistry with L8A4 antibody; magnification 400× (**A**) and Western blot (**B**) analysis), as well as on the mRNA level (real-time PCR results (**C**)). However, MLPA (Multiplex-ligation dependent probe amplification) analysis of DK-MG^extra-high^, DK-MG^high^, and DK-MG^low^ did not reveal any significant alterations in gene profiles, as can be seen in the graph (**D**).

**Figure 2 ijms-23-12129-f002:**
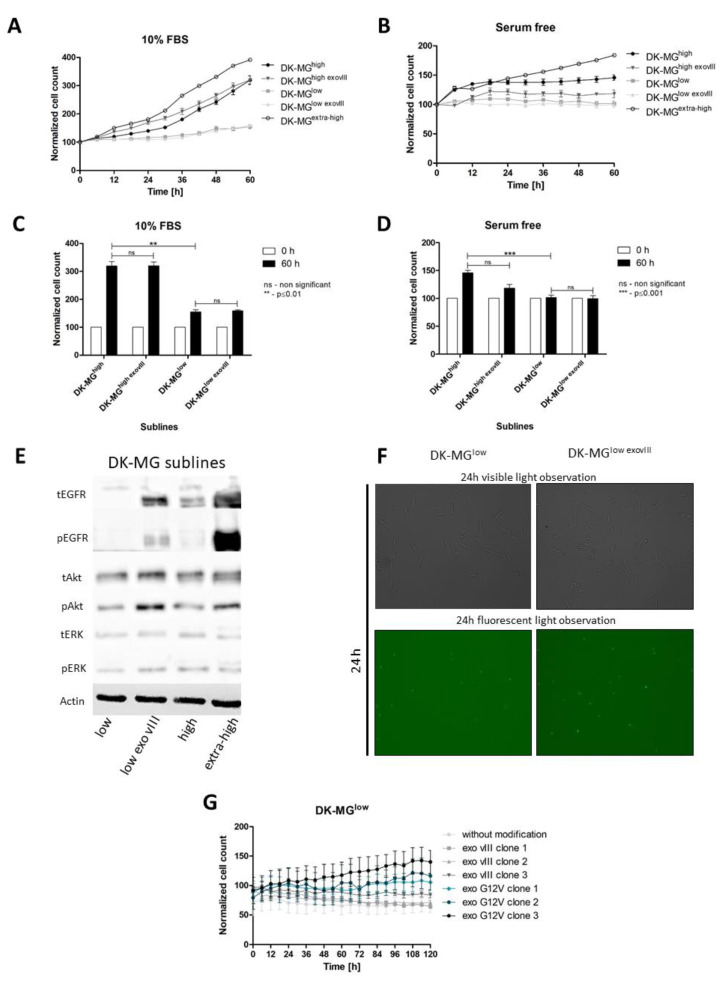
Influence of exogenous EGFR^vIII^ expression on the DK-MG subline status. Incorporation of EGFR^vIII^ into the DK-MG^high^ or DK-MG^low^ cell line did not significantly change their proliferation ratio in either full medium (**A**) or serum-free conditions (**B**) in statistically significant way (**C**,**D**). (**E**) The DK-MG^low^ cells regained an aggressive DK-MG^high^ phenotype following induction by exogenous K-RAS^G12V^ with no lack of rescue observed after EGFR^vIII^ transgene introduction. The lentiviral clones differed from each other by increasing virus titers for both EGFR^vIII^ and K-RAS^G12V^. No differences were found between DK-MG sublines with regard to the phosphorylation level of the downstream proteins in the EGFR transduction pathways; magnification 100× (**F**). Exogenous expression of EGFR^vIII^ in DK-MG^low^ was insufficient for avoiding spontaneous cell death (**G**).

**Figure 3 ijms-23-12129-f003:**
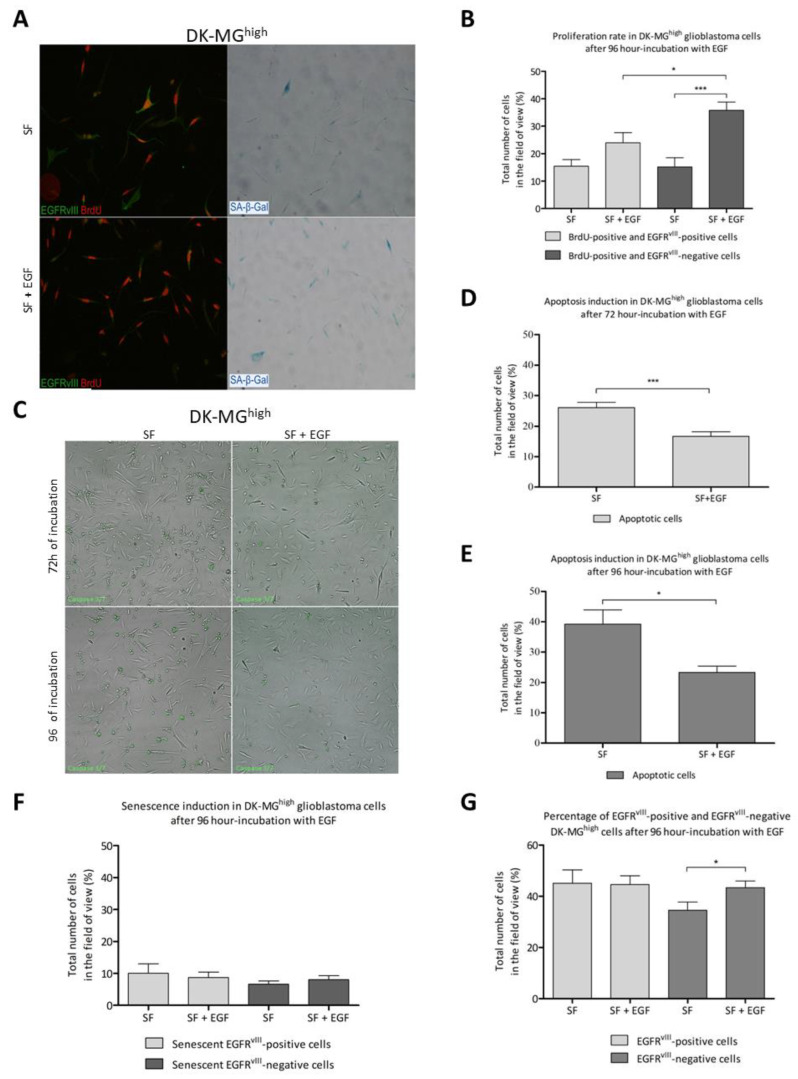
The influence of EGF on senescence induction, proliferation rate, apoptosis, and EGFR^vIII^ expression in DK-MG^high^ glioblastoma cells in vitro. Representative images showing the impact of EGF on cellular proliferation and senescence in DK-MG^high^ glioblastoma cells; magnification 400× (**A**). Comparison of the percentage of proliferating EGFR^vIII^-positive and EGFR^vIII^-negative cells after 96 h of incubation with 20 ng/mL EGF (SF (serum free)+ EGF) and without EGF (SF) (**B**). Representative images showing the EGF influence on apoptosis induction in DK-MG^high^ glioblastoma cells after 72 and 96 h of incubation with epidermal growth factor; magnification 40× (**C**). Comparison of the percentage of apoptotic glioblastoma cells after 72 h of incubation with 20 ng/mL EGF (SF + EGF) and w/o EGF (SF) (**D**). Comparative analysis of apoptosis induction in DK-MG^high^ cells in response to 96-h treatment with 20 ng/mL EGF (SF + EGF) and w/o EGF (SF) (**E**). Comparative analysis of senescence induction in both EGFR^vIII^-positive and EGFR^vIII^-negative DK-MG^high^ cells in response to 96-h treatment with 20 ng/mL EGF (SF + EGF) and w/o EGF (SF) (**F**). The percentage of EGFR^vIII^-positive and EGFR^vIII^-negative cells after 96-h incubation with EGF (SF + EGF) and w/o EGF (SF) (**G**). SF–serum free medium/serum-free conditions. Comparison 20× objective used. Statistical significance was considered at *p* < 0.05, “*”; *p* < 0.001, “***”.

**Figure 4 ijms-23-12129-f004:**
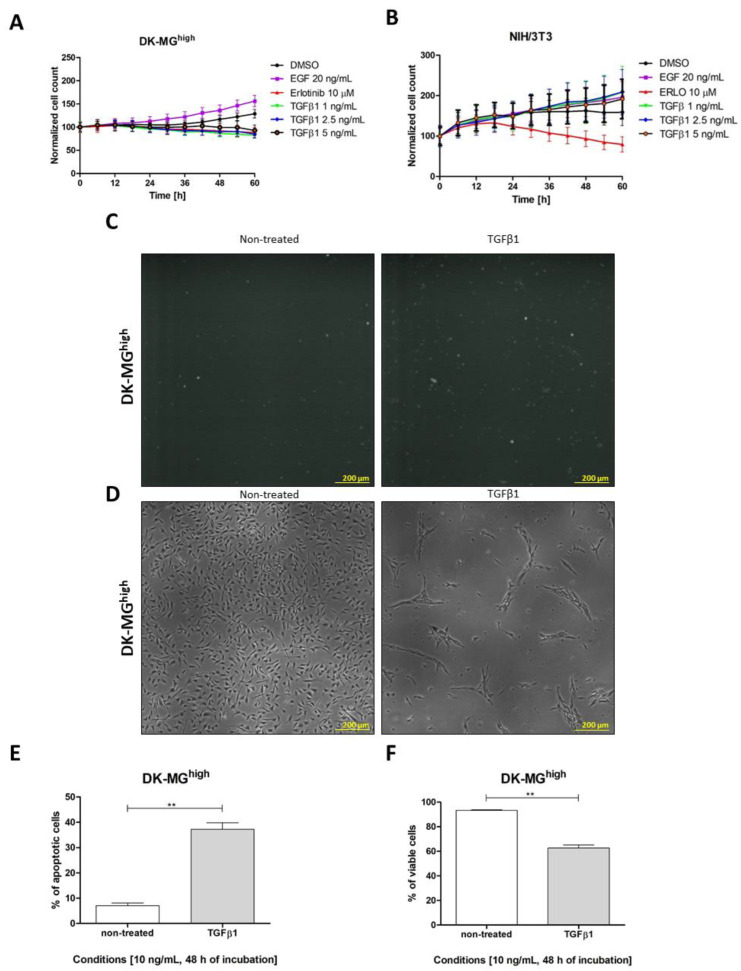
The influence of TGFβ1 on cell lines with different EGFR^vIII^ expressions. TGFβ1 was very effective in the DK-MG^high^ cell line, with cell death visible after 1–5 ng/mL TGFβ1 treatment in DK-MG^high^ (**A**), while no changes in cell viability or the proliferation ratio were found for NIH/3T3 fibroblasts (**B**). It is known that spontaneous cell death may occur in the DK-MG^high^ subline; however, after TGFβ1 treatment, the number of apoptotic cells (**C**) increased. On the other hand, the number of viable cells was higher in the non-treated control than in cells exposed to TGFβ1 (**D**). In both analyses, the observed differences were statistically significant (**E**,**F**). Statistical significance was considered at *p* < 0.01, “**”.

**Figure 5 ijms-23-12129-f005:**
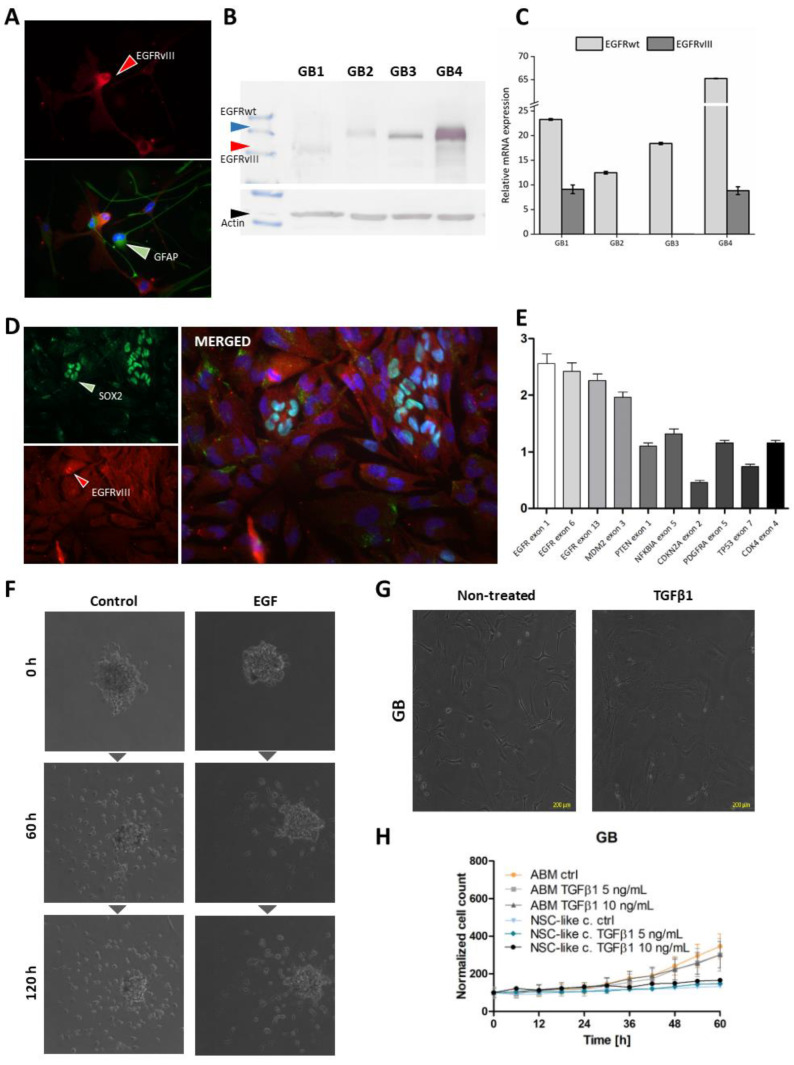
The influence of EGF and TGFβ1 on EGFR^vIII^-positive primary glioblastoma cell cultures. The characteristics of selected GB primary culture models was confirmed at the protein level by immunocytochemical co-staining of EGFRvIII with GFAP (magnification 400×) (**A**) and by Western blotting of EGFRvIII expression, with or without EGFRwt co-expression (**B**), as well as at the mRNA level by real-time qPCR (**C**). Co-staining of EGFRvIII and SOX-2 protein expression revealed both EGFRvIII-positive SOX-2-positive and EGFRvIII-positive SOX-2-negative GB cells in primary cultures; magnification 400× (**D**). MLPA for EGFRvIII-positive primary GBs (here GB4) was not interpretable due to EGFRwt co-amplification (**E**). Some spheres isolated from the same EGFRvIII-positive primary GB responded to EGF by proliferation while others deteriorated; magnification 100× (**F**). The influence of TGFβ1 on the primary GB cell line with confirmed EGFRvIII expression was analyzed. GB cell morphology was initially mixed between mesenchymal and epithelial; however, after 48 h of TGFβ1 treatment, this changed to a mesenchymal phenotype’ magnification 100× (**G**). The GB proliferation ratios in different cell culture conditions (ABM or NSC-like) after treatment with 5–10 ng/mL TGFβ demonstrated that the number of normal cells increased with the mesenchymal phenotype (**H**).

**Table 1 ijms-23-12129-t001:** NGS analysis of DK-MG sublines. Novel mutations detected by Ion Torrent™ Personal Genome Machine sequencing in DK-MG^low^ and DK-MG^-high^ sublines, which had not been previously published in the Cancer Cell Line Encyclopedia.

Cell Line	Gene ID	Type	Frequency %	Allele Call	Chromosome	Position	Reference	Variant	Coverage
Only DKMG low	CSF1R	MNP	99.1	Homozygous	5	149,433,596	TG	GA	106
Only DKMG low	APC	SNP	50.6	Heterozygous	5	112,175,770	G	A	915
DKMG low and high	KDR	SNP	100	Homozygous	4	55,980,239	C	T	495
DKMG low and high	RET	SNP	100	Homozygous	10	43,613,843	G	T	253
DKMG low and high	FLT3	SNP	100	Homozygous	13	28,610,183	A	G	459
DKMG low and high	TP53	SNP	52.8	Heterozygous	17	7,579,472	G	C	303

**Table 2 ijms-23-12129-t002:** Comparison of DK-MG sublines. EGFR^wt^ and EGFR^vIII^ expression was assessed by real-time PCR analysis in DK-MG sublines with endogenous (DK-MH^low^, DK-MG^high^, DK-MG^extra-high^) and exogenous EGFR^vIII^ expression (DK-MG^low exovIII^ and DK-MG^high exovIII^). The percentage of EGFR^vIII^-positive cells was estimated according to immunocytochemical staining performed on total EGFR and EGFR^vIII^. Doubling time was assessed by microscopy.

Cell Subline	EGFR Total DNA Copy Number	EGFR WT cDNA Relative Expression	EGFR vIII cDNA Relative Expression	% EGFR vIII Positive Cells	Doubling Time
DK-MG^low^	25	4.05	0.1	1–5%	44 h
DK-MG^high^	218	4.25	371	40–60%	23 h
DK-MG^extra-high^	196	3.80	289	100%	14 h
DK-MG^low exo vIII^	223	4.46	325	100%	43 h
DK-MG^high exo vIII^	226	4.53	382	100%	23 h

**Table 3 ijms-23-12129-t003:** The IC_50_ values for TGFβ1 for EGFR^vIII^-positive neoplastic cells (DK-MG^high^ and primary glioblastoma cultures) and normal cells (NIH/3T3 fibroblasts, pericytes, and NSCs (neural stem cells) following 72-h incubation (MTT (3-(4,5-Dimethylthiazol-2-yl)-2,5-Diphenyltetrazolium Bromide) assay).

Cell Line	DK-MG High	NIH/3T3	Pericytes	Neural Stem Cells	Primary GB1	Primary GB4	Primary GB7
IC50 value concentration range	4 ng/mL	1000 ng/mL	>1000 ng/mL	>1000 ng/mL	14 ng/mL	>1000 ng/mL	>1000 ng/mL

## Data Availability

All data generated or analyzed during this study are included in this published article and its Appendix A. P.R. or E.S.-F. should be contacted if someone wants to request the data.

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
