# Peer review of "Phenotypical Flexibility of the EGFRvIII-Positive Glioblastoma Cell Line and the Multidirectional Influence of TGFβ and EGF on These Cells—EGFRvIII Appears as a Weak Oncogene"

_ijms, 2022, doi:10.3390/ijms232012129_

Round 1

Reviewer 1 Report

 The authors extensively analyzed biological characteristics and phenotypes of EGFRvIII in DK-MG, which is a unique in vitro model of truncated EGFR, through various experimental techniques, even though the results were a little confusing. The manuscript properly shows logical flows of experimental details but also shows so many grammatical errors in English expressions, which need professional editing by native speakers or experts.

Page 2, line 64~67: poor English, difficult to understand. Please, re-write the sentences

Page 2, line 89: changed to “The in vitro senescence in glioblastoma cells is ~”

Page 5, line 148~153 & figure 1D: It looks like the bar graph in figure 1D was not matched for described text of DK-MG high & DK-MG extra-high sublines in manuscript. Please, check this point.

Figure 2A, 2B, 2D: no labels within the cell images, except 2C (Sa-beta-Gal)

Page 10, line 244~247: The author mentioned the cytostatic effect of TGFb was not observed in iNS in the manuscript. However, this point is not matched to Supplementary figure 2F (lower cell count of TGFb treated cells in iNS)

Page 13, line 319~321: This sentence is not understandable due to improper English expression.

Page 14, line 385~389: This sentence is not understandable due to improper English expression.

Reviewer 2 Report

This manuscript deals with the expression of EGFRvIII on glioblastoma cell line and the influence of EGF and TGFβ on the same cell line. Three glioblastoma DK-MG sublines: DK-18 MGlow, DK-MGhigh and DK-MGextra-high were tested with EGF and TGFβ.

More importantly, in the abstract, it is reported that primary glioblastoma cells analysis showed elimination of EGFRvIII positive cells when exposed to EGF and a process similar to EMT after exposure to TGFβ.

The authors conclude that the role of TGFβ and EGF in EGFRvIII context may be important by its novelty and could suggest new active context-dependent pathways in cancer cells that can be targeted. Also, the importance of EGFRvIII is still uncertain, questioning CAR-T targeting this protein.

The topic is of interest as the expression of the mutant form of EGF that is the EGFRvIII can be considered a target for therapy; indeed, the expression of this mutant form is somehow debated because not all GBM cells in a given patient express this molecule. More importantly, the function of this isoform is still to be defined. Moreover, GBM cells can express at the same time the non-mutated EGFR and so the effect of EGF on these GBM cells is of great interest.

This work is mainly focused on the effect of EGF and TGFbeta1 on the cell line DK-MG and its sublines with a different percentage of EGFRvIII+ cells.

 The data regarding the primary GBM cells are reported on page 8 lines 215-219 as follows:

 “EGF effect on primary GB cells was also tested. Primary glioblastoma cells cultured in several various monolayer conditions showed gradual increase in the percentage of senescent cells with every passage. It was presented several times that 3D conditions protect most efficiently GB cells against in vitro senescence [25].”

 Unfortunately, I cannot find any characterization of these primary cell lines, expression of  EGFRvIII+  and effect of EGF or TGFbeta1 in this manuscript. Probably this portion of the paper has been deleted or forgotten.

I think this is essential to further understand the relevance of EGFRvIII+. In our experimental experience, we did not find a good expression of EGFRvIII+ on primary GBM cells. Also, at very early passage of culture and with cells able to well proliferate to EGF. Actually, I do not think that EGFRvIII+ is relevant in the biology of GBM cells. So, I consider of great interest the analysis performed by the authors. But, in this present form it is limited to DK-MG sublines and the conclusion I can get from the data shown is simply that the EGFRvIII is not so essential. Also, the EGFRvIII expression can influence the biology and response to growth and differentiating factors of a single subline.

 In other words, the data regarding the effects found should be analysed on primary GBM cells, otherwise the manuscript is really limited to the data shown on a single subline of GBM. Also, the EGFRvIII+ cells in primary GBM should be studied to reinforce the data obtained with the single cell line analysed.

Furthermore, the silencing of EGFR can be obtained perhaps and see the function of the EGFRvIII+ cells.

Round 2

Reviewer 2 Report

The authors have replied to reviewer's query. The manuscript can be endorsed for publication.